# Solving Structured Hierarchical Games Using Differential Backward Induction*

**Zun Li**[1]    **Feiran Jia**[2]    **Aditya Mate**[3]    **Shahin Jabbari**[4]    **Mithun Chakraborty**[1]    **Milind Tambe**[3]

**Yevgeniy Vorobeychik**[5]

[1]University of Michigan, Ann Arbor, `{lizun,dcsmc}@umich.edu`
[2]Pennsylvania State University, `fzj5059@psu.edu`
[3]Harvard University, `{aditya_mate,milind_tambe}@g.harvard.edu`
[4]Drexel University, `shahin@drexel.edu`
[5]Washington University in St. Louis, `yvorobeychik@wustl.edu`

## Abstract

From large-scale organizations to decentralized political systems, hierarchical strategic decision making is commonplace. We introduce a novel class of *structured hierarchical games (SHGs)* that formally capture such hierarchical strategic interactions. In an SHG, each player is a node in a tree, and strategic choices of players are sequenced from root to leaves, with root moving first, followed by its children, then followed by their children, and so on until the leaves. A player's utility in an SHG depends on its own decision, and on the choices of its parent and *all* the tree leaves. SHGs thus generalize simultaneous-move games, as well as Stackelberg games with many followers. We leverage the structure of both the sequence of player moves as well as payoff dependence to develop a gradient-based back propagation-style algorithm, which we call *Differential Backward Induction (DBI)*, for approximating equilibria of SHGs. We provide a sufficient condition for convergence of DBI and demonstrate its efficacy in finding approximate equilibrium solutions to several SHG models of hierarchical policy-making problems.

## 1 INTRODUCTION

The COVID-19 pandemic has revealed considerable strategic tension among the many parties involved in decentralized hierarchical policy-making. For example, recommendations by the World Health Organization are sometimes heeded, and other times discarded by nations, while subnational units, such as provinces and urban areas, may in turn take a policy stance (such as on lockdowns, mask mandates, or vaccination priorities) that is not congruent with national

policies. Similarly, in the US, policy recommendations at the federal level can be implemented in a variety of ways by the states, while counties and cities, in turn, may comply with state-level policies, or not, potentially triggering litigation [15]. Central to all these cases is that, besides this strategic drama, what ultimately determines infection spread is how policies are implemented *at the lowest level*, such as by cities and towns, or even individuals. Similar strategic encounters routinely play out in large-scale organizations, where actions throughout the management hierarchy are ultimately reflected in the decisions made at the lowest level (e.g., by the employees who are ultimately involved in production), and these lowest-level decisions play a decisive role in the organizational welfare.

We propose a novel model of hierarchical decision making which is a natural stylized representation of strategic interactions of this kind. Our model, which we term *structured hierarchical games (SHGs)*, represents each player by a node in a tree hierarchy. The tree plays two roles in SHGs. First, it captures the sequence of moves by the players: the root (the lone member of level 1 of the hierarchy) makes the first strategic choice, its children (i.e., all nodes in level 2) observe the root's choice and follow, their children then follow in turn, and so on, until we reach the leaf node players who move upon observing their predecessors' choices. Second, the tree partially captures strategic dependence: a player's utility depends on its own strategy, that of its parent, and the strategies of *all of the leaf nodes*. The sequence of moves in our model naturally captures the typical sequence of decisions in hierarchical policy-making settings, as well as in large organizations, while the utility structure captures the decisive role of leaf nodes (e.g., individual compliance with vaccination policies), as well as hierarchical dependence (e.g., employee dependence on a manager's approval of their performance, or state dependence on federal funding). Significantly, the *SHG* model generalizes a number of well-established models of strategic encounters, including (a) simultaneous-move games (captured by a 2-level SHG with the root having a single dummy action), (b) Stackelberg

---

*The full technical version of this paper is available at `https://arxiv.org/abs/2106.04663`.

*Accepted for the 38th Conference on Uncertainty in Artificial Intelligence* (UAI 2022).

(leader-follower) games (a 2-level game with a single leaf node) [11, 32], and (c) single-leader multi-follower Stackelberg games (e.g., a Stackelberg security game with a single defender and many attackers) [5, 8].

Our second contribution is a gradient-based algorithm for approximately computing subgame-perfect equilibria of *SHGs*. Specifically, we propose *Differential Backward Induction (DBI)*, which is a backpropagation-style gradient ascent algorithm that leverages both the sequential structure of the game, as well as the utility structure of the players. As *DBI* involves simultaneous gradient updates of players in the same level (particularly at the leaves), convergence is not guaranteed in general (as is also the case for best-response dynamics [12]). Viewing *DBI* as a dynamical system, we provide a sufficient condition for its convergence to a stable point. Our results also imply that in the special case of two-player zero-sum Stackelberg games, *DBI* converges to a local Stackelberg equilibrium [11, 34].

Finally, we demonstrate the efficacy of DBI in finding approximate equilibrium solutions to several classes of SHGs. First, we use a highly stylized class of SHGs with polynomial utility functions to compare DBI with five baseline gradient-based approaches from prior literature. Second, we use DBI to solve a recently proposed game-theoretic model of 3-level hierarchical epidemic policy making. Third, we apply DBI to solve a hierarchical variant of a public goods game, which naturally captures the decentralization of decision making in public good investment decisions, such as investments in sustainable energy. Fourth, we evaluate DBI in the context of a hierarchical security investment game, where hierarchical decentralization (e.g., involving federal government, industry sectors, and particular organizations) can also play a crucial role. In all of these, we show that DBI significantly outperforms the state of the art approaches that can be applied to solve games with hierarchical structure.

**Related Work** SHGs generalize both simultaneous-move games and Stackelberg games with multiple followers [5, 21]. They are also related to *graphical games* [19] in capturing utility dependence structure, although SHGs also capture sequential structure of decisions. Several prior approaches use gradient-based methods for solving games with particular structure. A prominent example is generative adversarial networks (GANs), though these are zero-sum games [9, 14, 18, 26, 27, 28]. Ideas from learning GANs have been adopted in gradient-based approaches to solve multi-player general-sum games [4, 7, 16, 20, 23, 25, 26]. However, all of these approaches assume a simultaneous-move game. A closely-related thread to our work considers gradient-based methods for bi-level optimization [22, 31]. Several related efforts consider gradient-based learning in Stackelberg games, and also use the implicit function theorem to derive gradient updates [2, 11, 29, 33, 34]. We significantly generalize these ideas by considering an arbitrary hierarchical game structure.

Jia et al. [17] recently considered a stylized 3-level SHG for pandemic policy making, and proposed several non-gradient-based algorithms for this problem. We compare with their approach in Section 4.

## 2 STRUCTURED HIERARCHICAL GAMES

**Notation** We use bold lower-case letters to denote vectors. Let $f$ be a function of the form $f(\boldsymbol{x}, \boldsymbol{y}) : \mathbb{R}^d \times \mathbb{R}^{d'} \to \mathbb{R}^{d''}$. We use $\nabla_{\boldsymbol{x}} f$ to denote the partial derivative of $f$ with respect to $\boldsymbol{x}$. When there is functional dependency between $\boldsymbol{x}$ and $\boldsymbol{y}$, we use $D_{\boldsymbol{x}} f$ to denote the total derivative of $f(\boldsymbol{x}, \boldsymbol{y}(\boldsymbol{x}))$ with respect to $\boldsymbol{x}$. We use $\nabla_{\boldsymbol{x},\boldsymbol{x}}^2 f$ and $\nabla_{\boldsymbol{x},\boldsymbol{y}}^2 f$ to denote the second-order partial derivatives and $D_{\boldsymbol{x},\boldsymbol{x}}^2 f$ to denote the second-order total derivative of $f$. For a mapping $f : \mathbb{R}^d \to \mathbb{R}^d$, we use $f^t(\boldsymbol{x})$ to denote $t$ iterative applications of $f$ on $\boldsymbol{x}$. For mappings $f_1 : \mathbb{R}^d \to \mathbb{R}^d$ and $f_2 : \mathbb{R}^d \to \mathbb{R}^d$, we define $(f_1 \circ f_2)(\boldsymbol{x}) \triangleq f_1(f_2(\boldsymbol{x}))$ and $(f_1 + f_2)(\boldsymbol{x}) \triangleq f_1(\boldsymbol{x}) + f_2(\boldsymbol{x})$. Moreover, for a given $\epsilon \in \mathbb{R}^{>0}$ and $\boldsymbol{x} \in \mathbb{R}^d$, we define the $\epsilon$-ball around $\boldsymbol{x}$ as $\mathbb{B}_\epsilon(\boldsymbol{x}) = \{\boldsymbol{x}' \in \mathbb{R}^d \mid \|\boldsymbol{x} - \boldsymbol{x}'\|_2 < \epsilon\}$. Finally, $\boldsymbol{I}$ denotes an identity matrix.

**Formal Model** A structured hierarchical game (SHG) $\mathcal{G}$ consists of the set $\mathcal{N}$ of $n$ players. Each player $i$ is associated with a set of actions $\mathcal{X}_i \subseteq \mathbb{R}^{d_i}$. The players are partitioned across $L$ levels, where $\mathcal{N}_l$ is the set of $n_l$ players occupying level $l$. Let $l_i$ denote the level occupied by player $i$. This *hierarchical* structure of the game is illustrated in Figure 1 where players correspond to nodes and levels are marked by dashed boundaries. The hierarchy plays two crucial roles: 1) it determines the order of moves, and 2) it partly determines utility dependence among players. Specifically, the temporal pattern of actions is as follows: level 1 has a single player, the *root*, who chooses an action first, followed by all players in level 2 making simultaneous choices, followed in turn by players in level 3, and so on until the *leaves* in the final level $L$. Players of level $l$ only observe the actions chosen by all players of levels $1, 2, ..., l-1$, but not their peers in the same level. So, for example, pandemic social distancing and vaccination policies in the US are initiated by the federal government (including the Centers for Disease Control and Prevention who acts as the root in our game model), with states (second level nodes) subsequently instituting their own policies, counties (third level nodes) reacting to these by determining their own, and behavior of people (leaf nodes) ultimately influenced, but not determined, by the guidelines and enforcement policies by the local county/city.

Next, we describe the utility structure of the game as entailed by the SHG hierarchy. Each player $i$ in level $l_i > 1$ (i.e., any node other than the root) has a *unique parent* in level $l_i - 1$; we denote the parent of node $i$ by $\text{PA}(i)$. A player's utility function is determined by 1) its own action, 2) the action of

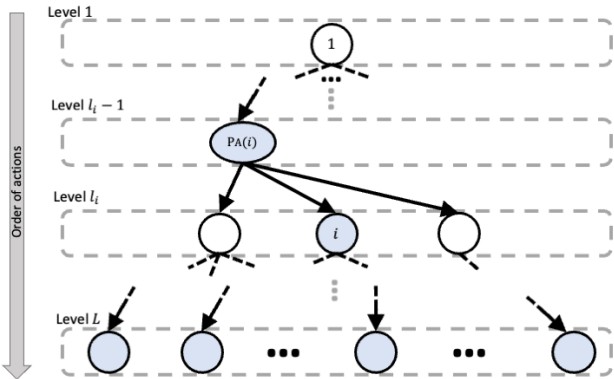

Figure 1: Schematic representation of an SHG. The utility of player $i$ can have direct functional dependence *only* on the joint action of *all* shaded players.

its parent, and 3) the actions of *all* players in level $L$ (i.e., all *leaf* players). To formalize, let $\boldsymbol{x}_l$ denote the joint action profile of all players in level $l$. Player $i$'s utility function then has the form $u_i(x_i, \boldsymbol{x}_L)$ if $l_i = 1$, $u_i(x_i, x_{\text{PA}(i)}, \boldsymbol{x}_L)$ if $1 < l_i < L$, and $u_i(x_i, x_{\text{PA}(i)}, \boldsymbol{x}_{L,-i})$ if $l_i = L$, where $\boldsymbol{x}_{L,-i}$ is the action profile of all players in level $L$ other than $i$. For example, in our running pandemic policy example, the utility of a county depends on both the policy and enforcement strategy of its state (its *parent*) and on the ultimate pandemic spread and economic impact within it, both determined largely by the behavior of the county residents (*leaf nodes*). Note the considerable generality of the SHG model. For example, an arbitrary simultaneous-move game is a SHG with 2 levels and a "dummy" root node (utilities of all leaves depend on one another's actions), and an arbitrary Stackelberg game (e.g., Stackelberg security game), even with many followers, can be modeled as a 2-level SHG with the leader as root and followers as leaves. Furthermore, while we have defined SHGs with respect to real-vector player action sets, it is straightforward to represent mixed strategies of finite-action games in this way by simply using a softmax function to map an arbitrary real vector into a valid mixed strategy.

**Solution Concept** Since an SHG has important sequential structure, it is natural to consider the *subgame perfect equilibrium (SPE)* as the solution concept [30]. Here, we focus on pure-strategy equilibria. To begin, we note that in SHGs, the strategies of players in any level $l > 1$ are, in general, functions of the complete history of play in levels $1, \ldots, l-1$, which we denote by $h_{<l} = (\boldsymbol{x}_1, \boldsymbol{x}_2, \ldots, \boldsymbol{x}_{l-1})$. Formally, a (pure) strategy of a player $i$ is denoted by $s_i(h_{<l})$, which deterministically maps an arbitrary history $h_{<l}$ into an action $x_i \in \mathcal{X}_i$. A *Nash equilibrium* of an SHG is then a strategy profile $\boldsymbol{s} = (s_1, \ldots, s_i, \ldots, s_n)$ such that for all $i \in \mathcal{N}$, $u_i(s_i, \boldsymbol{s}_{-i}) \geqslant u_i(s_i', \boldsymbol{s}_{-i})$ for all possible alternative strategies for $i$, $s_i'$. Here, we denote the realized payoff of $i$ from profile $\boldsymbol{s}$ by $u_i(s_i, \boldsymbol{s}_{-i})$. Next, we define a *level-$l$-subgame* given $h_{<l}$ as an SHG that includes only

players at levels $\geqslant l$, with actions chosen in levels $< l$ fixed to $h_{<l}$. A strategy profile $\boldsymbol{s}$ is a *subgame perfect equilibrium* of SHG if it is a Nash equilibrium of every level-$l$-subgame of SHG for every $l$ and history $h_{<l}$. We prove in the long version that our definition of SPE is equivalent to the standard SPE in an extensive-form representation of SHG.

While in principle we can compute an SPE of an SHG using backward induction, this cannot be done directly (i.e., by complete enumeration of actions of all players) as actions are real vectors. Moreover, even discretizing actions is of little help, as the hierarchical nature of the game leads to exponential explosion of the search space. We now present a gradient-based approach for approximating SPE along the equilibrium path in an SHG that leverages the game structure to derive backpropagation-style gradient updates.

# 3 DIFFERENTIAL BACKWARD INDUCTION

In this section, we describe our gradient-based algorithm, Differential Backward Induction (DBI), for approximating an SPE (which we mean hereinafter finding a joint-action profile $\boldsymbol{x}$ that constitutes a subgame-perfect equilibrium path), and then analyze its convergence. Just as gradient ascent does not, in general, identify a globally optimal solution to a non-convex optimization problem, DBI in general yields a solution which only satisfies first-order conditions (see Section 3.2 for further details). Moreover, we leverage the structure of the utility functions to focus computation on an SPE in which strategies of players are only a function of their immediate parents.[1]

In this spirit, we define *local* best response functions $\phi_i : \mathbb{R}^{d_{\text{PA}(i)}} \rightarrow \mathbb{R}^{d_i}$ mapping a player $i$'s parent's action $x_{\text{PA}(i)}$ to $i$'s action $x_i$; note that the notation $\phi_i$ is distinct from $s_i$ above for $i$'s strategy to emphasize the fact that $\phi_i$ is only locally optimal. Now, suppose that a player $i$ is in the last level $L$. Local optimality of $\phi_i$ implies that if $x_i = \phi_i(x_{\text{PA}(i)})$, then $\nabla_{x_i} u_i\left(\boldsymbol{x}_i, \boldsymbol{x}_{\text{PA}(i)}, \boldsymbol{x}_{L,-i}\right) = 0$ and $\nabla^2_{x_i,x_i} u_i\left(\boldsymbol{x}_i, \boldsymbol{x}_{\text{PA}(i)}, \boldsymbol{x}_{L,-i}\right) < 0$.[2]

Let $\phi_l$ denote the local best response for all the players in level $l$ given the actions of all players in level $l-1$. We can compose these local best response functions to define the function $\Phi_l := \phi_L \circ \phi_{L-1} \circ \ldots \circ \phi_{l+1} : \mathbb{R}^{d_{n_l}} \rightarrow \mathbb{R}^{d_{n_L}}$ i.e., the local best response of players in the last level $L$ given the actions of the players in level $l$.[3] Then for any player $(i)$ in level $l_i < L$, $D_{x_i} u_i\left(x_i, x_{\text{PA}(i)}, \Phi_l\left(\langle x_i, \boldsymbol{x}_{l,-i}\rangle\right)\right) = 0$

---

[1]Note that while we cannot guarantee that an SPE exists in SHGs in general, let alone those possessing the assumed structure, we find experimentally that our approach often yields good SPE approximations.

[2]For simplicity, we omit degenerate cases where $\nabla^2_{\boldsymbol{x}_i,\boldsymbol{x}_i} u_i = 0$ and assume all local maxima are strict.

[3]Note that in particular $\Phi_L = \phi_L$.

and $D^2_{x_i,x_i} u_i \left( x_i, x_{\mathrm{PA}(i)}, \Phi_l \left( \langle x_i, \boldsymbol{x}_{l,-i} \rangle \right) \right) \prec 0$, where $D_{x_i}$ is the total derivative with respect to $x_i$ (as $\Phi_l(\langle x_i, \boldsymbol{x}_{l,-i} \rangle)$ is also a function of $x_i$). Note that the functions $\phi$ and $\Phi$ are *implicit*, capturing the functional dependencies between actions of players in different levels at the local equilibrium.

Throughout, we make the following standard assumption on the utility functions [10, 34].

**Assumption 1.** *For any $x_i \in \mathcal{X}_i$, the second-order partial derivatives of the form $\nabla^2_{x_i,x_i} u_i$ are non-singular.*

## 3.1 ALGORITHM

The DBI algorithm works in a bottom-up manner, akin to back-propagation: for each level $l$, we compute the total derivatives (gradients) of the utility functions and local best response maps $(\phi, \Phi)$ based on analytical expressions that we derive below. We then propagate this information up to level $l - 1$, as it is used to compute gradients for that level, and so on until level 1. Algorithm 1 gives the full DBI algorithm. In this algorithm, $\mathrm{CHD}(i)$ denotes the set of children of player $i$ (i.e., nodes in level $l_i + 1$ for whom $i$ is the parent). DBI works in a backward message-passing

---

**Algorithm 1** Differential Backward Induction (DBI)
___
**Input:** An SHG instance $\mathcal{G}$
**Parameters:** Learning rate $\alpha$, maximum number of iterations $T$ for gradient update
**Output:** A strategy profile
  Randomly initialize $\boldsymbol{x}^0 = \langle \boldsymbol{x}_1^0, \ldots, \boldsymbol{x}_L^0 \rangle$
  **for** $t = 1, 2, \ldots, T$ **do**
    **for** $l = L, L-1, \ldots, 1$ **do**
      **for** $i = 1, 2, \ldots, n_l$ **do**
        **if** $l = L$ **then**
          Back-propagate $D_{x_i} \Phi_i = \boldsymbol{I}$ to $\mathrm{PA}(i)$
          Set $x_i^t \leftarrow x_i^{t-1} + \alpha \nabla_{x_i} u_i$
        **else**
          Compute $\nabla_{x_i} u_i, \nabla_{\boldsymbol{x}_L} u_i$ at $\boldsymbol{x}^{t-1}$
          Compute $D_{x_i} \phi_j, \forall j \in \mathrm{CHD}(i)$ (Eqn. (5))
          Compute $D_{x_i} \Phi_l$ (Eqn. (4))
          Back-propagate $D_{x_i} \Phi_l$ to $\mathrm{PA}(i)$
          Compute $D_{x_i} u_i = \nabla_{x_i} u_i + \nabla_{\boldsymbol{x}_L} u_i D_{x_i} \Phi_l$
          Set $x_i^t \leftarrow x_i^{t-1} + \alpha D_{x_i} u_i$
  Return $\boldsymbol{x}^T$
___

manner, comparable to back-propagation: after each player has computed its total derivative, it passes (back-propagates) $D_{x_i} \Phi_l$ to its direct parent; this information is, in turn, used by the parent to compute its own total derivative, which is passed to its own parent, and so on.

Algorithm 1 takes the total derivates as given. We now derive closed-form expressions for these. We start from the last level $L$. Given the actions of players in level $L - 1$, the total

derivative of a player $i \in \mathcal{N}_L$ with respect to $x_i$ is

$$D_{x_i} u_i \left( x_i, x_{\mathrm{PA}(i)}, \boldsymbol{x}_{L,-i} \right) = \nabla_{x_i} u_i. \quad (1)$$

For a player $i$ in level $L - 1$, the total derivative (at a local best response) is

$$D_{x_i} u_i(x_i, x_{\mathrm{PA}(i)}, \phi_L(\langle x_i, \boldsymbol{x}_{L-1,-i} \rangle))$$
$$= \nabla_{x_i} u_i + (\nabla_{\boldsymbol{x}_L} u_i)(D_{x_i} \phi_L), \quad (2)$$

where $\nabla_{\boldsymbol{x}_L} u_i$ is a $1 \times d_{n_L}$ vector and $D_{x_i} \phi_L$ is a $d_{n_L} \times d$ matrix. The technical challenge here is to derive the term $D_{x_i} \phi_L$ for $i \in \mathcal{N}_{L-1}$. Recall that $\phi_L$ is the vectorized concatenation of the $\phi_j$ functions for $j \in \mathcal{N}_L$. Since the local best response strategy of a player in level $L$ only depends on its parent in level $L - 1$, the only terms in $\phi_L$ that depend on $x_i$ are the actions of $\mathrm{CHD}(i)$ in level $L$. Consequently, it suffices to derive $D_{x_i} \phi_j$ for $j \in \mathrm{CHD}(i)$. Note that for these players $j$, $\nabla_{x_j} u_j = 0$ (by local optimality of $\phi_L$). We will use this first-order condition to derive the expression for the total derivative using the *implicit function theorem*.

**Theorem 1** (Implicit Function Theorem (IFT) [10, Theorem 1B.1]). *. Let $f(\boldsymbol{x}_1, \boldsymbol{x}_2) : \mathbb{R}^d \times \mathbb{R}^d \to \mathbb{R}^d$ be a continuously differentiable function in a neighborhood of $(\boldsymbol{x}_1^*, \boldsymbol{x}_2^*)$ such that $f(\boldsymbol{x}_1^*, \boldsymbol{x}_2^*) = 0$. Also suppose $\nabla_{\boldsymbol{x}_2} f$, the Jacobian of $f$ with respect to $\boldsymbol{x}_2$, is non-singular at $(\boldsymbol{x}_1^*, \boldsymbol{x}_2^*)$. Then around a neighborhood of $\boldsymbol{x}_1^*$, we have a local diffeomorphism $\boldsymbol{x}_2^*(\boldsymbol{x}_1) : \mathbb{R}^d \to \mathbb{R}^d$ such that $D_{\boldsymbol{x}_1} \boldsymbol{x}_2 = - (\nabla_{\boldsymbol{x}_2} f)^{-1} \nabla_{\boldsymbol{x}_1} f$.*

To use Theorem 1, we set $f = \nabla_{x_j} u_j$ (which satisfies the conditions of Theorem 1 by Assumption 1), $\boldsymbol{x}_1 = x_i$ and $x_2 = \boldsymbol{x}_j$ (recall that $j \in \mathrm{CHD}(i)$). By IFT, there exists $\phi_j(x_i)$ such that $D_{x_i} \phi_j = -(\nabla^2_{x_j,x_j} u_j)^{-1} \nabla^2_{x_j,x_i} u_j$. Define $\nabla^2_j := \nabla^2_{x_j,x_i} u_j$. Then

$$(\nabla_{\boldsymbol{x}_L} u_i)(D_{x_i} \phi_L) = -\sum_{j \in \mathrm{CHD}(i)} (\nabla_{x_j} u_i) D_{x_i} \phi_j$$
$$= -\sum_{j \in \mathrm{CHD}(i)} (\nabla_{x_j} u_i)(\nabla^2_{x_j,x_j} u_j)^{-1} \nabla^2_j.$$

Plugging this into Equation (2), we obtain

$$D_{x_i} u_i \left( x_i, x_{\mathrm{PA}(i)}, \phi_L \left( \boldsymbol{x}_{L-1} \right) \right)$$
$$= \nabla_{x_i} u_i - \sum_{j \in \mathrm{CHD}(i)} (\nabla_{x_j} u_i)(\nabla^2_{x_j,x_j} u_j)^{-1} \nabla^2_j.$$
$$(3)$$

For a level $l < L-1$, the total derivative of player $i \in \mathcal{N}_l$ in a local best response is $D_{x_i} u_i = \nabla_{x_i} u_i + (\nabla_{\boldsymbol{x}_L} u_i)(D_{x_i} \Phi_l)$, where

$$D_{x_i} \Phi_l = \left( D_{\boldsymbol{x}_{l+1}} \Phi_{l+1} \right) \left( D_{x_i} \boldsymbol{x}_{l+1} \right)$$

$$= \sum_{j \in \text{CHD}(i)} \left( D_{x_j} \Phi_{l+1} \right) \left( D_{x_i} \phi_j \right). \qquad (4)$$

Applying IFT, we get

$$D_{x_i} \phi_j = -(\nabla^2_{x_j, x_j} u_j)^{-1} \nabla^2_{x_j, x_i} u_j, \qquad (5)$$

for $j \in \text{CHD}(i)$. We can apply the above procedure recursively for $D_{\boldsymbol{x}_{l+1}} \Phi_{l+1}$ to derive the total derivative for players $i \in \mathcal{N}_l$ for $l < L - 1$:

$$D_{x_i} u_i = \nabla_{x_i} u_i + \left( \sum_{j \in \text{LEAF}(i)} (-1)^{L-l} \nabla_{x_j} u_i \right.$$
$$\left. \prod_{\eta \in \text{PATH}(j \to i)} \left( \nabla^2_{x_\eta, x_\eta} u_\eta \right)^{-1} \nabla^2_{x_\eta, \boldsymbol{x}_{\text{PA}(\eta)}} u_\eta \right),$$
$$(6)$$

where $\text{PATH}(j \to i)$ is an ordered list of nodes (players) lying on the unique path from $j$ to $i$, excluding $i$. Note that Equation (6) is a generalization of Equation (3) where the PATH only consists of the leaf player.

While the above derivation assumes the $\phi$ and $\Phi$ functions are local best responses, in our algorithm in each iteration we evaluate these functional expressions for the total derivatives *at the current joint action profile*. This significantly reduces computational complexity and ensures that Algorithm 1 satisfies the first-order conditions upon convergence.

## 3.2 CONVERGENCE ANALYSIS

As we remarked earlier, stable points of DBI are not guaranteed to be SPE just as stable points of gradient ascent are not guaranteed to be globally optimal with general non-convex objective functions. Furthermore, DBI algorithm entails what are effectively iterative better-response updates by players, and it is well-known that best response dynamic processes in games will in general lead to cycles [25].

In spite of these challenges, we provide sufficient conditions for the DBI algorithm to converge to a stable point. In particular, in the rest of this section, we first show that the gradient updates of DBI can be written as a dynamical system and characterize the conditions in which this system will converge to an stable point (Proposition 1). We then show how DBI can be tuned (in terms of learning rate in Proposition 2, number of iterations in Proposition 3 and initializations in Proposition 4) to converge to such stable points when they exists. While the set of stable points and approximate SPEs are not necessarily the same, we empirically show that DBI is effective in converging to SPEs.

To begin, we observe that the gradient updates in DBI can be interpreted as a discrete dynamical system, $\boldsymbol{x}^{t+1} = F(\boldsymbol{x}^t)$, with $F(\boldsymbol{x}^t) = (\boldsymbol{I} + \alpha G)(\boldsymbol{x}^t)$ where $G$ is an update gradient

vector. This discrete system can be viewed as an approximation of a continuous limit dynamical system $\dot{\boldsymbol{x}} = G(\boldsymbol{x})$ (i.e., letting $\alpha \to 0$). A standard solution concept for such dynamical systems is a *locally asymptotic stable point (LASP)*.

**Definition 1** ([13])**.** *A continuous (or discrete) dynamical system $\dot{\boldsymbol{x}} = G(\boldsymbol{x})$ (or $\boldsymbol{x}^{t+1} = F(\boldsymbol{x}^t)$) has a locally asymptotic stable point (LASP) $\boldsymbol{x}^*$ if $\exists \epsilon > 0, \lim_{t \to \infty} \boldsymbol{x}^t = \boldsymbol{x}^*, \forall \boldsymbol{x}^0 \in \mathbb{B}_\epsilon(\boldsymbol{x}^*)$.*

There are well-known necessary and sufficient conditions for the existence of an LASP.

**Proposition 1** (Characterization of LASP [35, Theorem 1.2.5, Theorem 3.2.1])**.** *A point $\boldsymbol{x}^*$ is an LASP for the continuous dynamical system $\dot{\boldsymbol{x}} = G(\boldsymbol{x})$ if $G(\boldsymbol{x}^*) = 0$ and all eigenvalues of Jacobian matrix $\nabla_{\boldsymbol{x}} G$ at $\boldsymbol{x}^*$ have negative real parts. Furthermore, for any $\boldsymbol{x}^*$ such that $G(\boldsymbol{x}^*) = 0$, if $\nabla_{\boldsymbol{x}} G$ has eigenvalues with positive real parts at $\boldsymbol{x}^*$, then $\boldsymbol{x}^*$ cannot be an LASP.*

Note that an LASP of DBI is an action profile of all players that satisfies the first-order conditions, i.e., it has the property that no player can improve their utility through a local gradient update. While the existence of an LASP depends on game structure, we show that under Assumption 1, and as long as the sufficient conditions for LASP existence in Proposition 1 are satisfied, DBI converges to LASP. We defer all the omitted proofs to the long version.

**Proposition 2.** *Let $\lambda_1, \dots, \lambda_d$ denote the eigenvalues of the updating Jacobian $\nabla_{\boldsymbol{x}} G$ at an LASP $\boldsymbol{x}^*$ and define $\lambda^* = \arg\max_{i \in [d]} Re(\lambda_i)/|\lambda_i|^2$, where $Re$ is the real part operator. Then with a learning rate $\alpha < -2Re(\lambda^*)/|\lambda^*|^2$, and an initial point $\boldsymbol{x}^0 \in \mathbb{B}_\epsilon(\boldsymbol{x}^*)$ for some $\epsilon > 0$ around $\boldsymbol{x}^*$, DBI converges to an LASP. Specifically, if the choice of learning rate equals $\alpha^*$ and the modulus of matrix $\rho(\boldsymbol{I} + \alpha^* \nabla_{\boldsymbol{x}} G) = 1 - \kappa < 1$, then the dynamics converge to $\boldsymbol{x}^*$ with the rate of $O((1 - \kappa/2)^t)$.*

Proposition 2 states that there exists a region such that, if the initial point is in that region, then DBI will converge to an LASP. We next show that if we assume first-order Lipschitzness for the update rule, then we can also characterize the region of initial points which converge to an LASP.

**Proposition 3.** *Suppose $G$ is L-Lipschitz.[4] Then for all $\boldsymbol{x}^0 \in \mathbb{B}_{\kappa/2L}(\boldsymbol{x}^*)$, $\epsilon > 0$ and after $T$ rounds of gradient update, DBI will output a point $\boldsymbol{x}^T \in \mathbb{B}_\epsilon(\boldsymbol{x}^*)$ as long as $T \geq \lceil \frac{2}{\kappa} \log \|\boldsymbol{x}^0 - \boldsymbol{x}^*\| / \epsilon \rceil$ where $\kappa$ is as defined in Proposition 2.*

We further show that through random initialization, the probability of reaching a *saddle point* is 0, which means that

---

[4]Formally, this means that $\exists L > 0$ such that $\forall \boldsymbol{x}, \boldsymbol{x}' \in \mathcal{X}, \|G(\boldsymbol{x}) - G(\boldsymbol{x}')\|_2 \leq L \|\boldsymbol{x} - \boldsymbol{x}'\|_2$.

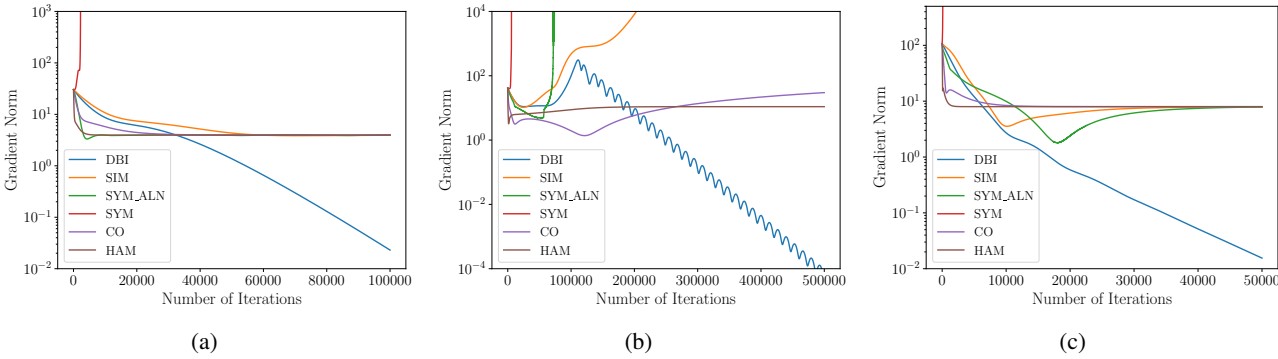

Figure 2: Convergence behaviors on (a) a $(1, 1, 1)$ game with 1-d actions (b) a $(1, 1, 2)$ game with 1-d actions (c) a $(1, 1, 1)$ game with 3-d actions.

with probability 1, DBI converges to an LASP in which players are playing *local* best responses.

**Proposition 4.** *Suppose $G$ is L-Lipschitz. Let $\alpha < 1/L$ and define the saddle points of the dynamics $G$ as $\mathcal{X}^*_{sad} = \{x^* \in \mathcal{X} \mid x^* = (I + \alpha G)(x^*), \rho((I + \alpha \nabla_x G)(x^*)) > 1\}$. Also let $\mathcal{X}^0_{sad} = \{x^0 \in \mathcal{X} \mid \lim_{t \to \infty} (I + \alpha G)^t(x^0) \in \mathcal{X}^*_{sad}\}$ denote the set of initial points that converge to a saddle point. Then $\mu(\mathcal{X}^0_{sad}) = 0$, where $\mu$ is Lebesgue measure.*

While our convergence analysis does not guarantee convergence to an approximate SPE, our experiments show that DBI is in fact quite effective in doing so in practice.

## 4 EXPERIMENTS

In this section, we empirically investigate the following questions: (1) the convergence rate of DBI, (2) the solution quality of DBI, (3) the behavior of DBI in games where we can verify global stability. All our code is written in python. We ran our experiments on an Intel(R) Core(TM) i7-7700HQ CPU @ 2.80GHz to obtain the results in Sections 4.1, and on an Intel(R) Core(TM) i9-9820X CPU @ 3.30GHz for the rest of the experiments. [5]

We evaluate the performance in terms of quality of equilibrium approximation as a function of the number of iterations of a given algorithm, or its running time. Ideally, given a collection of actions $x$ played by players along the (approximate) equilibrium path computed, we wish to find the largest utility gain any player can have by deviating from this path, which we denote by $\epsilon(x)$. However, this computation is impossible in our setting, as it would need to consider all possible histories as well, whereas our approach and alternatives only return $x$ along the path of play (moreover, considering all possible histories is itself intractable).

[5]Code available at https://github.com/jtongxin/SHG_DBI.

Therefore, we consider two heuristic alternatives. The first, which we call *local SPE regret*, runs DBI for every player $i$ starting with $x$, and returns the greatest benefit that any player can thereby obtain; we use this in Section 4.1. In the rest of this section, we use the second alternative, which we call *global SPE regret*. It considers for each player $i$ in level $l$ a discrete grid of alternative actions, and uses best response dynamics to compute an approximate SPE of the level-$(l + 1)$ subgame to evaluate player $i$'s utility for each such deviation. This approach then returns the highest regret among all players computed in this way.

Our evaluation considers three SHG scenarios. We begin by comparing DBI to a number of baselines on simple, stylized SHG models, then move on to three complex hierarchical game models motivated by concrete applications.

### 4.1 POLYNOMIAL GAMES

We begin by considering instances of SHGs to which we can readily apply several state-of-the-art baselines, allowing us a direct comparison to previous work. Specifically, we consider 3 SHG instances with different game properties: (a) a three-level chain structure (or the $(1, 1, 1)$ game) with 1-d actions (b) a "⋋" shape tree (or the $(1, 1, 2)$ game) with 1-d action spaces, and (c) and $(1, 1, 1)$ game with 3-d actions. In all the games, the payoffs are polynomial functions of $x$ with randomly generated coefficients (we can think of these as proxies for a Taylor series approximation of actual utility functions). The exact coefficient of these polynomial functions as well as an analysis of the running time of each method can be found in the long version.

We compare DBI with the following five baselines: 1) simultaneous partial gradient ascent (SIM) [7, 25], 2) symplectic gradient dynamics with or 3) without alignment (SYM_ALN and SYM, respectively) [4], 4) consensus optimization (CO) [27], and 5) Hamilton gradient (HAM) [1, 24]. SIM, SYM_ALN, SYM, CO and HAM are

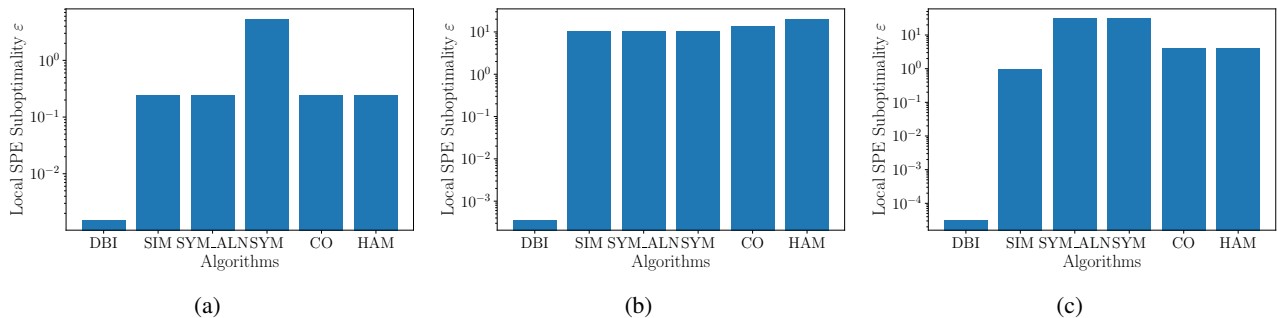

Figure 3: Solution qualities on (a) a $(1, 1, 1)$ game with 1-d actions (b) a $(1, 1, 2)$ game with 1-d actions (c) a $(1, 1, 1)$ game with 3-d actions.

all designed to compute a local Nash equilibrium [4, 7].

We start by comparing convergence behavior of DBI to the baselines. We run all algorithm with the same initial point and learning rate. The results are in Figure 2 where we plot the $L_2$ norm of total gradient for each of the algorithms (Y axis) against the number of iterations (X axis).

In all cases, DBI converges to a critical point that meets the first-order conditions while the baseline algorithms fail to do so in most cases. In Figures 2(a) and (c), all baselines have converged to a point with finite norm for the total gradients. In (b), however, only CO and HAM converge to a stationary point while SIM, SYM, SYM_ALN all diverge. For scenario (b), DBI appears to be on an inward spiral to a critical point. We further check the second-order condition (see the long version) and verify that DBI has actually converged to local maxima of individual payoffs in all three games.

Next, we investigate solution quality in terms of *local regret* of DBI compared to baselines. As shown in Figure 3, across all three game instances, DBI outputs a profile of actions (along the path of play) with near-zero local regret while other algorithm fail to do so.

## 4.2 DECENTRALIZED EPIDEMIC POLICY GAME

Next, we consider DBI for solving a class of games inspired by hierarchical decentralized policy-making in the context of epidemics such as COVID-19 [17]. The hierarchy has levels corresponding to the (single) federal government, multiple states, and county administrations under each state. Each player's action (policy) is a scalar in $[0, 1]$ that represents, for example, the extent of social distancing recommended or mandated by a player (e.g., a state) for its administrative subordinates (e.g., counties). Crucially, these subordinates have considerable autonomy about setting their own policies, but incur a non-compliance cost for significantly deviating from recommendations made by the level immediately above (of course, non-compliance costs are not relevant for the root

player). The full cost function of each player additionally includes an infection prevalence within the geographic territory of interest to the associated entity (e.g., within the state), as well as the socio-economic cost of the policy itself. To summarize, the total cost for each player is a combination of the infection cost, socio-economic cost as well as the non-compliance cost (when applicable). However, different players can have different combinations of these cost (through player-specific weights for each of the costs) that can lead to strategic tensions between the players (see the long version for details).

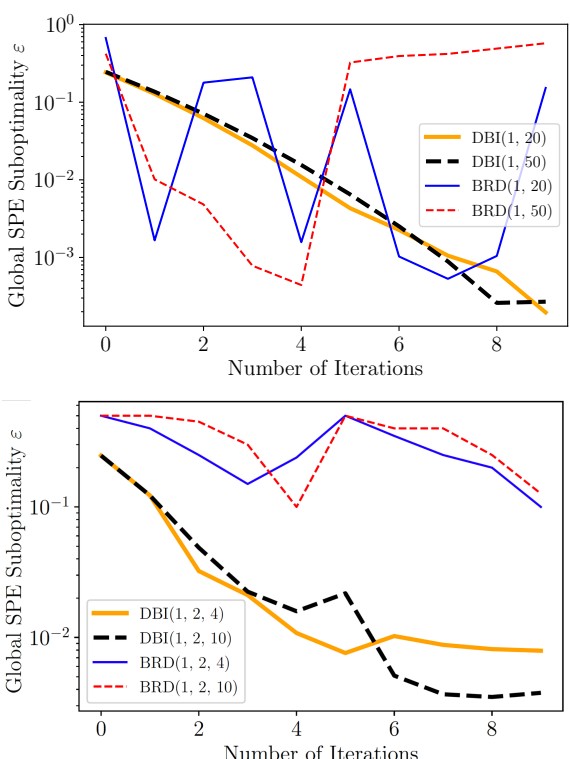

Figure 4: Global regret for the decentralized epidemic policy game. Top and bottom columns correspond to result for games with 2 and 3 levels, respectively.

Since the actions are in a one-dimensional compact space and the depth of the hierarchy is at most 3, our baseline is the best response dynamics (BRD) algorithm proposed by Jia et al. [17] (detailed in the long version), and we use *global regret* as a measure of efficacy in comparing the proposed DBI algorithm with BRD. The results of this comparison are shown in Figures 4 and 5 for two-level (government and states) and three-level (government, states, counties) variants of this game. We consider two-level games with 20 and 50 leaves (states), and three-level games with 2 players in level 2 (states) and 4 and 10 leaves (counties).

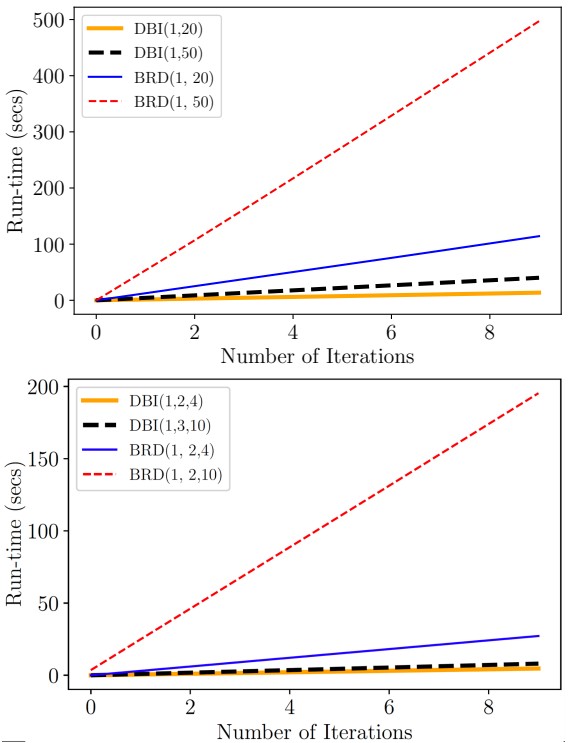

Figure 5: Running time for the decentralized epidemic policy game. Top and bottom columns correspond to result for games with 2 and 3 levels, respectively.

As we can see in Figure 4, BRD can have poor convergence behavior in terms of global regret, whereas DBI appears to converge quite reliably to a path of play with a considerably lower global regret. Notably, the improvement in solution quality becomes more substantial as we increase the game complexity either in terms of scale (number of leaves) or in terms of the level of hierarchy (moving from 2- to 3-level games).

Running time (in seconds) demonstrates the relative efficacy of DBI even further (see Figure 5). In particular, observe the significant increase in the running time of BRD as we increase the number of leaves. In contrast, DBI is far more scalable: indeed, even more than doubling the number of players appears to have little impact on its running time. Moreover, BRD is several orders of magnitude slower than

DBI for the more complex games.

## 4.3 HIERARCHICAL PUBLIC GOODS GAMES

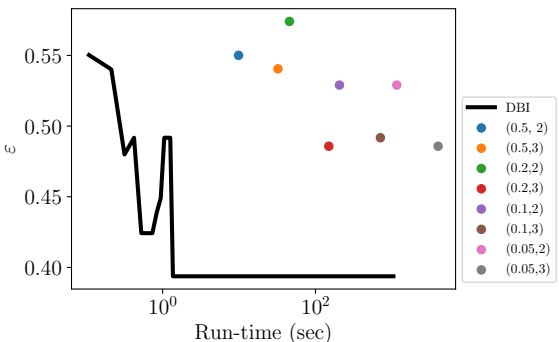

Figure 6: Performance ($\epsilon$) in the Public Goods Game; the scatter points show the results of BRD with discretization factors $0.5, 0.2, 0.1, 0.05$, and best response rounds $2, 3$.

Next, we consider *hierarchical public goods games*. A conventional networked public goods game endows each player $i$ with a utility function $u_i(x_i, x_{-i}) = a_i + b_i x_i + \sum_j g_{ji} x_i x_j - c_i(x_i)$, where $g_{ji}$ is the impact of player $j$ on player $i$ (often represented as a weighted edge on a network), and $x_i \in [0, 1]$ the level of investment in the public good by player $i$ [6]. We construct a 3-level hierarchical variant of such games by starting with the karate club network [36] which represents friendships among 34 individuals. Level-2 nodes are obtained by partitioning the network into two (sub)clubs, with leaves (level-3 nodes) representing all the individuals. The utility of level-2 nodes is the sum of utilities of individual members of associated clubs, with the utility of the root being the sum of the utilities of all individuals. Furthermore, we introduce non-compliance costs with investment policies in the level immediately above, as we did in the decentralized epidemic policy game (Section 4.2). Further details on the exact form of the utility functions and parameters of the games are provided in in the long version.

Figure 6 presents the global regret as a function of running time for DBI (black line) and BRD with different levels of discretization (dots). We observe that DBI yields considerably lower regret in these games than BRD even as we discretize the latter finely. Moreover, DBI reaches smaller regret orders of magnitude faster than BRD.

## 4.4 HIERARCHICAL SECURITY GAMES

In the final set of experiments, we evaluate DBI on a hierarchical extension of *interdependent security games* [3]. In these games, $n$ defenders can each invest $x_i \geqslant 0$ in security. If defender $i$ is attacked, the probability that the attack succeeds is $1/(1 + x_i)$. Furthermore, defenders are interdependent, so that a successful attack on defender $i$ cascades

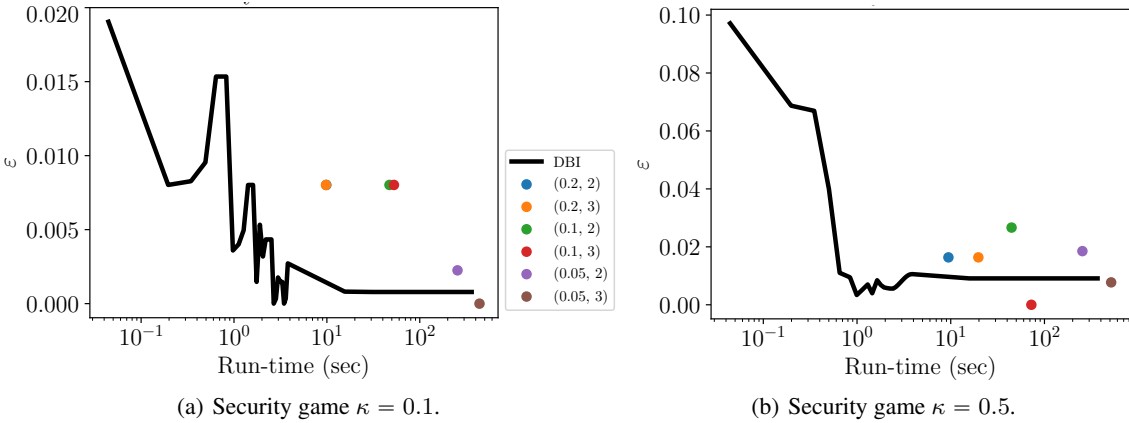

(a) Security game $\kappa = 0.1$.

(b) Security game $\kappa = 0.5$.

Figure 7: Results on $(1, 3, 6)$ hierarchical security games. (a) $\kappa = 0.1$ and (b) $\kappa = 0.5$; legend is shared.

to defender $j$ with probability $q_{ji}$. In the variant we adopt, the attacker strategy is a uniform distribution over defenders (e.g., the "attacker" is just nature, with attacks representing stochastic exogenous failures). The utility of the defender is the probability of surviving the attack less the cost of security investment.

We extend this simultaneous-move game to a hierarchical structure consisting of one root player (e.g., government), three level-2 players (e.g., sectors), and six leaf players (e.g., organizations). The policy-makers in the first two levels of the game recommend an investment policy to the level below, and aim to maximize total welfare (sum of utilities) among the leaf players in their subtrees. Just as in both hierarchical epidemic and public goods games, whenever a player in level $l$ does not act according to the recommendation of their parent in level $l-1$, they incur a non-compliance cost. Complete model details are deferred to the long version. We conduct experiments with two weights $\kappa$ that determine the relative importance of non-compliance costs in the decisions of non-root players in the game: $\kappa \in \{0.1, 0.5\}$.

Figures 7(a) and 7(b) present the results of comparing DBI with BRD on this class of games, where BRD is again evaluated with different levels of action space discretization (note, moreover, that in this setting discretizing actions is not enough, since these are unbounded, and we also had to impose an upper bound). We can observe that for either value of $\kappa$, DBI yields high-quality SPE approximation (in terms of global SPE regret) far more quickly than BRD. In particular, when we use relatively coarse discretization, BRD is approximately an order of magnitude slower, and yields significantly higher regret. In contrast, if we use finer discretization for BRD, global regret for BRD and DBI becomes comparable, but now BRD is several orders of magnitude slower. For example, DBI converges within several seconds, whereas if we discretize $x_i$ into multiples of 0.02, BRD takes nearly 2 hours, while discretization at the level of 0.01 results in BRD taking nearly 7 hours.

## 5   CONCLUSION

We introduced a novel class of hierarchical games, proposed a new game-theoretic solution concept and designed an algorithm to compute it. We assume a specific form of utility dependency between players and our solution concept only guarantees local stability. Improvement on each of these two fronts is an interesting direction for future work.

Given the generality of our framework, our approach can be used for many applications characterized by a hierarchy of strategic agents e.g., pandemic policy making. However, our modeling requires the full knowledge of the true utility functions of all players and our analysis assumes full rationality for all the players. Although the model we have addressed here is already challenging, these assumptions are unlikely to hold in many real-world applications. Therefore, further analysis is necessary to fully gauge the robustness of our approach before deployment.

## ACKNOWLEDGMENTS

This work was supported in part by the US Army Research Office under MURI grant # W911NF-18-1-0208.

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
