# OpenReview forum: "Solving Structured Hierarchical Games Using Differential Backward Induction"
_auai.org/UAI/2022/Conference — UAI 2022 Oral_

### Official Review · Reviewer_gUMb · 2022-04-11

**Q2(1) Originality/Novelty:** 3
**Q2(2) Significance/Impact:** 3
**Q2(3) Correctness/Technical Quality:** 3
**Q2(6) Clarity Of Writing:** 3
**Q6 Overall Score:** 7
**Q8 Confidence In Your Score:** 3

**Q1 Summary And Contributions:**

The paper presents Structured Hierarchical Games (SHG), a new framework for strategic decision making involving a collection of players. These kinds of games provide a generalisation of simultaneous-move games as well as Stackelberg games with multiple followers. A new gradient-based back propagation algorithm called Differential Backward Induction (DBI) is introduced to solve SHGs efficiently. The empirical evaluation supports the claims of the paper.


**Q2 Assessment Of The Paper:**

More detailed information regarding each of these aspects is given below:

**Q2(4) Quality Of Experiments (Optional):**

4: Excellent: The experimental evaluation is comprehensive and the results are compelling.

**Q2(5) Reproducibility:**

3: Good: Key resources (e.g., proofs, code, data) are available and key details (e.g., proofs, experimental setup) are sufficiently well-described for competent researchers to confidently reproduce the main results.

**Q3 Main Strengths:**

1. The quality of presentation is overall fairly good. The paper is well organised and the material is presented in a relatively clear manner.

2. The empirical evaluation is fairly comprehensive, it includes a mix of both synthetic as well as more realistic multi-player games, and analyses the performance of the proposed approach in terms of running time as well as solution quality.

3. The paper considers an important class of strategic games that could have many real applications involving multiple decision makers.


**Q4 Main Weakness:**

Perhaps the main weakness of the paper is that it doesn't include a good running example that would support and help understand the technical details introduced throughout the paper.


**Q5 Detailed Comments To The Authors:**

Regarding the presentation, I found Figure 1 a bit too high-level. Maybe it would be possible to add a more concrete example such as one based on the COVID policy game presented in the experimental section.


**Q7 Justification For Your Score:**

The overall presentation is fairly good and the experiments appear to support and validate the main claims of the paper.


**Q9 Complying With Reviewing Instructions:**

1: Yes.

---

### Official Review · Reviewer_T1Nc · 2022-04-12

**Q2(1) Originality/Novelty:** 3
**Q2(2) Significance/Impact:** 2
**Q2(3) Correctness/Technical Quality:** 3
**Q2(6) Clarity Of Writing:** 3
**Q6 Overall Score:** 6
**Q8 Confidence In Your Score:** 2

**Q1 Summary And Contributions:**

Tye authors introduce structured hierarchical games. Where a player is exactly on one layer and only makes one decision.
The authors give a back-propagation algorithm that computes equilibria efficiently.
Said convergence time is provided by experiments.

**Q2 Assessment Of The Paper:**

More detailed information regarding each of these aspects is given below:

**Q2(4) Quality Of Experiments (Optional):**

3: Good: The experimental evaluation is adequate, and the results convincingly support the main claims.

**Q2(5) Reproducibility:**

3: Good: Key resources (e.g., proofs, code, data) are available and key details (e.g., proofs, experimental setup) are sufficiently well-described for competent researchers to confidently reproduce the main results.

**Q3 Main Strengths:**

Interesting model
The experiments are compelling

**Q4 Main Weakness:**

Perhaps look at more settings when showing the convergence time?


**Q5 Detailed Comments To The Authors:**

What would happen if a player appears multiple times in the tree?

**Q7 Justification For Your Score:**

This paper is firmly outside my field of expertise.

**Q9 Complying With Reviewing Instructions:**

1: Yes.

---

### Official Review · Reviewer_LPoR · 2022-04-12

**Q2(1) Originality/Novelty:** 3
**Q2(2) Significance/Impact:** 3
**Q2(3) Correctness/Technical Quality:** 3
**Q2(6) Clarity Of Writing:** 3
**Q6 Overall Score:** 5
**Q8 Confidence In Your Score:** 3

**Q1 Summary And Contributions:**

The authors introduce a class of structured hierarchical games (SHGs) and offer a heuristic algorithm for computing solutions of SHGs. The new class of games is quite expressive, as argued by the authors, and nicely captures such issues as pandemic policy making. The authors supplement their definitions and algorithm with extensive experiments.

**Q2 Assessment Of The Paper:**

More detailed information regarding each of these aspects is given below:

**Q2(4) Quality Of Experiments (Optional):**

2: Fair: The experimental evaluation is weak: important baselines are missing, or the results do not adequately support the main claims.

**Q2(5) Reproducibility:**

2: Fair: Key resources (e.g., proofs, code, data) are unavailable but key details (e.g., proof sketches, experimental setup) are sufficiently well-described for an expert to confidently reproduce the main results.

**Q3 Main Strengths:**

1) The new class of games is expressive, intuitive, and appealing

2) The experimental results suggest that the proposed algorithm is of value

3) The experiments are quite convincing in the sense that the authors model natural phenomena in a (semi) realistic way.

**Q4 Main Weakness:**

1) The experiments are missing huge amounts of detail. For example, in Section 4.2. the authors do not really give any technical information regarding the utility functions. While I understand that it takes a lot of space (indeed, the details are in the appendix), not having them in the main body of the paper makes it almost impossible to assess the experiments' value (in the "strengths" I wrote the experiments are convincing, by assuming their intuitive explanations of the experiments and utility functions are accurate, but I could not verify if they are).

2) Section 3.2. contains several propositions, but I am missing a summarizing theorem. Are the results really so minor that no theorem is warranted? (This is a minor criticism though)

**Q5 Detailed Comments To The Authors:**

The authors overuse the word "novel". I think that this word is justified if one offers a truly qualitative difference as compared to the state of the art. What the authors provide are "new" results and models, but certainly not "novel" ones.

I think that it would benefit the paper to give fewer experiments, but describe them in more detail.

**Q7 Justification For Your Score:**

The new class of games is appealing, the algorithm for solving them seems useful, but the experiments are not described sufficiently clearly.

**Q9 Complying With Reviewing Instructions:**

1: Yes.

---

### Decision · Program_Chairs · 2022-05-15

**Decision:**

Accept (Oral)

**Comment:**

Meta Review: The reviewers agreed this paper is worthy of acceptance. Please include a running example to make it more readably, and carry out the other suggestions as you agreed in the responses.